# Renewable energy production will exacerbate mining threats to biodiversity

Laura J. Sonter ⓘ [1,2,3 ✉], Marie C. Dade[1,2,4], James E. M. Watson ⓘ [1,2,5] & Rick K. Valenta ⓘ [6]

Renewable energy production is necessary to halt climate change and reverse associated biodiversity losses. However, generating the required technologies and infrastructure will drive an increase in the production of many metals, creating new mining threats for biodiversity. Here, we map mining areas and assess their spatial coincidence with biodiversity conservation sites and priorities. Mining potentially influences 50 million km$^2$ of Earth's land surface, with 8% coinciding with Protected Areas, 7% with Key Biodiversity Areas, and 16% with Remaining Wilderness. Most mining areas (82%) target materials needed for renewable energy production, and areas that overlap with Protected Areas and Remaining Wilderness contain a greater density of mines (our indicator of threat severity) compared to the overlapping mining areas that target other materials. Mining threats to biodiversity will increase as more mines target materials for renewable energy production and, without strategic planning, these new threats to biodiversity may surpass those averted by climate change mitigation.

[1] School of Earth and Environmental Sciences, The University of Queensland, Brisbane, QLD, Australia. [2] Centre for Biodiversity and Conservation Science, The University of Queensland, Brisbane, QLD, Australia. [3] Gund Institute for Environment, University of Vermont, Burlington, VT, USA. [4] Department of Geography, McGill University, Montreal, QC, Canada. [5] Wildlife Conservation Society, Bronx, NY, USA. [6] Sustainable Minerals Institute, The University of Queensland, Brisbane, QLD, Australia. ✉email: l.sonter@uq.edu.au

Climate change poses serious threats to biodiversity[1,2]. To keep temperature increases below 2 °C, and halt associated biodiversity losses, 140 nations committed to the Paris Climate Change Accord to reduce anthropogenic greenhouse gas emissions by 90% (from 2010 levels) and reach carbon neutrality by 2100[3]. Energy sector innovation is where most progress is achievable[4], but since renewable energies currently account for only 17% of global energy consumption[5], significant production increases must occur to phase out fossil fuel use[6]. However, the production of renewable energies is also material-intensive—much more so than fossil fuels[7]—meaning that future production will also escalate demand for many metals[8–11]. It is unlikely that these new demands will be met by diverting use from other sectors or from recycling materials alone[12,13]. When required commodities exist in biodiverse countries that lack strong resource governance, such as the world's second largest untouched lithium reserve in Bolivia's Salar de Uyuni salt pan[5]—a biodiverse area currently untouched by mining[14]—mining poses serious threats to species and ecosystems[15].

Global conservation efforts are often naive to the threats posed by significant growth in renewable energies. Production infrastructure (e.g. for wind and solar farms) has a significant spatial footprint[16] and other environmental risks[17], but potentially more extensive are the direct and indirect consequences of associated mining activities[15,18]. While some protected areas (PAs) prevent mineral extraction and prospecting activities, more than 14% of PAs contain metal mines within or nearby their boundaries[19] and consequences for biodiversity may extend many kilometers from mining sites[20]. Other areas that are increasingly important for future conservation investment (such as Key Biodiversity Areas[21]), are not designed to consider the distribution of mineral resources and pressures to extract them, as they focus instead on the needs of biodiversity only. Conservation plans for these sites must identify and develop strategies to manage all major threats to biodiversity, to ensure that mining the materials needed for renewable energy production does not simply replace the climate change-related threats mitigated by reducing fossil fuel use[22,23].

Here, we map the global extent of areas potentially influenced by mining, according to 62,381 pre-operational, operational, and closed mining properties that target 40 commodities (Supplementary Table 1). We use a 50 km wide radius around each mining property to capture the potential influence of both direct and indirect impacts of mining on biodiversity, but also produced a conservative estimate of mining areas using a 10 km radius (see "Methods" section). We distinguish areas targeting the materials that, among many other uses in society today, will be critical for renewable energy production (here referred to as critical mining areas) from those targeting other materials (e.g. fossil fuels and fertilizers; referred to as other mining areas). We quantify the spatial overlap of mining areas with nationally designated PAs[24] and sites considered important priorities for halting biodiversity loss (Key Biodiversity Areas[21] and Remaining Wilderness[25]). We compare this overlap between mining and non-mining areas (using the latter as a baseline); between critical and other mining areas (to determine threats by renewable energy production); and among closed, operational and pre-operational areas (to indicate potential future trends). We also develop and examine an index of mine density (i.e. the number of mining properties within 50 km of each 1 km² cell), which indicates local human pressure[26] and thus potential extinction risk for many endangered species[27], within overlap areas to determine differences between critical mining areas and other mining areas. We find that mining areas overlap with conservation areas and priorities and, although these areas are not more likely to overlap than other mining areas are, their areas overlapping with PAs and Remaining Wilderness do contain a greater mining density.

## Results and discussion

**Global extent and composition of mining areas.** Mining potentially influences 49.9 million km² of Earth's terrestrial land area (37%, excluding Antarctica), assuming impacts extend 50 km from mine sites (Fig. 1)—an enormous spatial footprint not specifically factored into global biodiversity threat maps[26] or conservation plans[28]. Most mining areas (82%) target materials critical for renewable energy production (Supplementary Table 2). Critical mining areas contain five times more mines and target three times more commodities than other mining areas do (Supplementary Table 1). However, despite their larger spatial extent, critical mining areas are less dense than other mining areas (2.0 vs. 3.5 for operational mining properties;

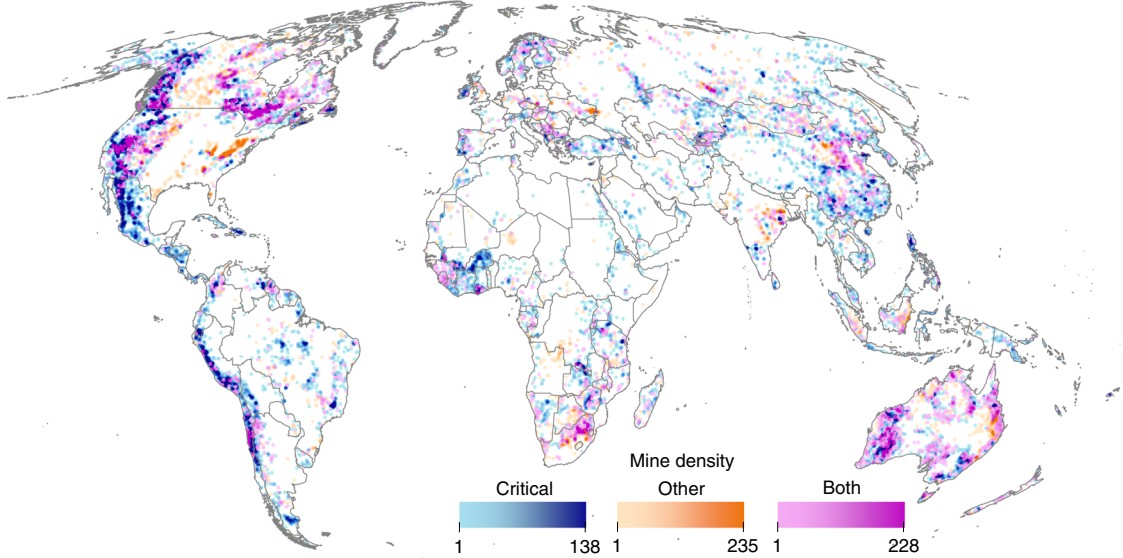

**Fig. 1 Global mining areas and their density.** Mining areas were mapped using a 50-cell radius around 62,381 pre-operational, operational, and closed mining properties. Mining areas with properties targeting materials critical for renewable energy technology and infrastructure are shown in blue, areas with properties targeting other materials are shown in orange, and those targeting both commodity types are shown in pink. Color shading (light to dark) indicates the density of mining areas—i.e. the number of mining properties within a 50-cell radius of each 1 km cell.

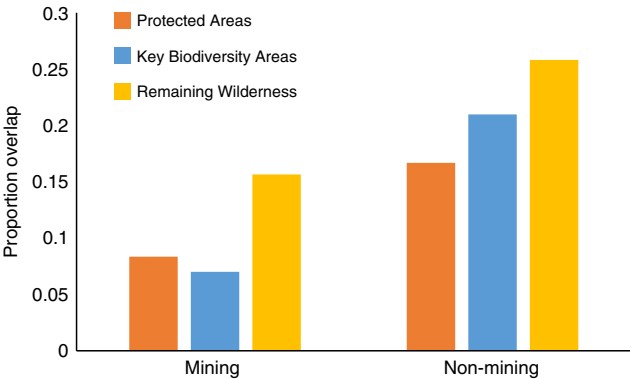

**Fig. 2 Overlap between mining and biodiversity conservation.** Bars depict the areal proportion of mining and non-mining areas that overlap with currently Protected Areas and conservation priorities (Key Biodiversity Areas and Remaining Wilderness). Mining areas were mapped using a 50-cell radius around known mining properties; non-mining areas include the remaining terrestrial land surface excluding Antarctica. Mining areas overlapped with Protected Areas and conservation priorities significantly less often than non-mining areas overlapped with Protected Areas and conservation priorities (Protected Areas: $n = 2$, $p < 0.001$; Key Biodiversity Areas: $n = 2$, $p = 0.009$; Remaining Wilderness: $n = 2$, $p < 0.001$). Results hold when comparing areas by mine status (i.e. pre-operating, operating, closed), except for closed regions overlapping with Key Biodiversity Areas, which did not differ significantly between mining and non-mining areas ($n = 2$, $p = 0.140$). Results also hold when mapping mining areas using a 10-cell radius around mining properties and when sampling the data at 300 km intervals. *Note:* Graphed proportions were calculated from full extent of mining and non-mining areas, whereas reported statistics were calculated using the $1\,km^2$ cells sampled at 100-cell intervals.

Supplementary Table 2), due to differences in resource distributions and relative production costs. For example, annual coal production (7813 Mt in 2018)[29] greatly exceeds that of copper metal (21 Mt)[30] (i.e. a commodity required for renewable energies; Table 1). Most copper is produced in the form of concentrate, so the tonnage of material transported in a similar way to coal would be in the order of 80 Mt. Thus, the world coal industry transports two orders of magnitude (100×) more material than the copper industry. This assertion is supported by production data per mine[31], showing that the average material transported from a coal mine is more than 10 times the amount of concentrate shipped from a copper mine. As a result, costs per mine for coal are more dependent on pre-existing proximate processing and transportation infrastructure, leading to a closer clustering of these operations.

**Biodiversity conservation sites and priorities within mining areas.** Approximately 8% of the global area potentially influenced by mining overlapped with PAs (Fig. 2), indicating extensive threats to currently protected biodiversity. However, these threats were significantly less than expected when compared to the overlap between non-mining areas and PAs (8.4% vs. 16.7%; chi-squared = 28.13, $p < 0.001$; Supplementary Data 1), possibly suggesting PAs do prevent some mining activities within their boundaries. It is also possible that mineral exploration occurs less often within PAs and this, in turn, lowers the rate of mining in these areas. We found less overlap between pre-operational mining areas (where minerals are not yet confirmed economically viable) and PAs than non-mining areas and PAs (12.2%; chi-squared = 55.54, $p < 0.001$; Supplementary Data 1), although pre-operational mining density did not differ significantly between areas overlapping PAs and areas not overlapping PAs (4.0 vs.

4.4 mining properties; Supplementary Data 2). Alternatively, it is possible that PAs are sited away from mineral resources or that, discovering minerals within PAs, has led to their abolishment. Since evidence does exist for mineral resources leading to world-wide downgrading, downsizing, and degazettement of PAs (PADDD)[32], further reducing mining threats to PAs, by strengthening their governance and management, is crucial if conservation efforts are to secure their biodiversity[33].

Mining areas also overlapped with conservation priority sites: 16% with Remaining Wilderness and 7% with Key Biodiversity Areas (Fig. 2). Similar to PAs, these overlap values were significantly less than those for non-mining areas (Remaining Wilderness: 25.8%, chi-squared = 115.36, $p < 0.001$; Key Biodiversity Areas: 21.0%, chi-squared = 6.82, $p = 0.009$; Supplementary Data 1). Note these differences were not statistically significant when using a 10-cell radius to define mining areas, or when sampling at 300 km intervals, but proportional overlap with mining areas was still less than non-mining areas in all cases. However, their extent of overlap was still substantial, particularly for areas containing pre-operational mines (Supplementary Data 1). While mineral prospecting does not (yet) cause the same threats to biodiversity as operational mining sites do, they are not without risk[15]. Exploration activities require new infrastructure to access remote areas, habitat clearing for seismic surveys and drilling, and can result in land and water contamination[34]. Observed differences between Remaining Wilderness and Key Biodiversity Areas (i.e. we found more Key Biodiversity Areas than Remaining Wilderness overlapped with mining areas) likely reflected differences in the metrics used to identify these sites. Remaining Wilderness may indirectly capture some mining threats, as sites with road networks and infrastructure are not considered wild thus resulting in low levels of overlap[25]; however, KBA designation completely ignores mines and mineral resources, focusing instead on where biodiversity occurs[21].

**Differences between critical and other mining areas.** We found no significant difference between critical and other mining areas in terms of their overlap with PAs (12.1% vs. 10.1%; chi-squared = 2.20, $p = 0.14$) or Key Biodiversity Areas (7.6% vs. 6.9%; chi-squared = 0.31, $p = 0.57$; Fig. 3; Supplementary Data 1). However, critical mining areas overlapping PAs were significantly more dense than the other mining areas that overlapped PAs (3.5 vs. 1.6; $D = 0.18$, $p = 0.01$; Supplementary Data 3; although differences were not statistically significant when using a 10-cell radius to define mining areas, or when sampling at 300 km intervals), despite other mining areas being (on average) more dense globally (Supplementary Table 1). We found no significant difference in mining density between critical and other mining areas overlapping Key Biodiversity Areas (2.9 vs. 4.6; $D = 0.18$, $p = 0.08$; Supplementary Data 3). Results suggest that an expansion in mining areas globally will threaten PAs and Key Biodiversity Areas (regardless of commodity type), but increasing the proportion of mines targeting materials critical for renewable energy production may also disproportionality increase the threats to biodiversity within currently protected areas. However, the ultimate impacts to biodiversity will depend on the mix of technologies used[35], their mineral needs and methods used to mine them[10,11], and the effectiveness of efforts to manage their environmental impacts.

In contrast, critical mining areas overlapped with Remaining Wilderness significantly less often than other mining areas did (14.3% vs. 18.8%; chi-squared = 9.93, $p = 0.001$; Supplementary Data 1), although differences emerged among areas containing mines of different statuses (Fig. 3). We found no difference in

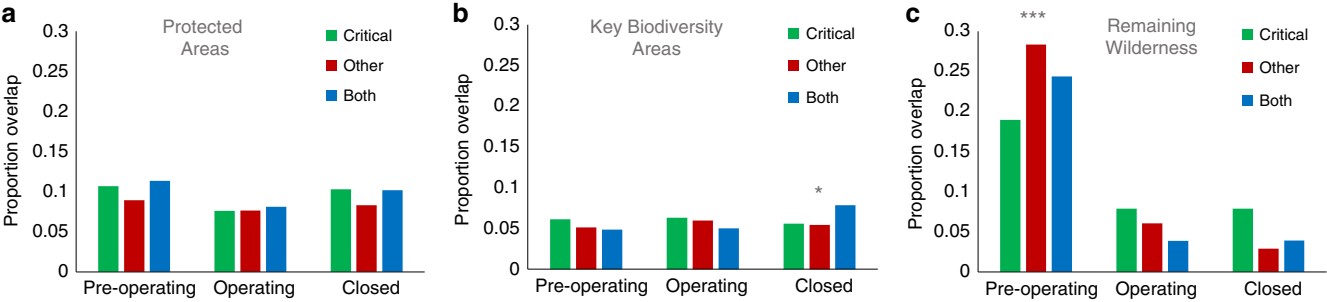

**Fig. 3 Overlap between mining and biodiversity conservation among critical, other and both mining areas.** Bars depict the areal proportion of critical, other, and both mining areas that overlap with currently Protected Areas (panel **a**) and conservation priorities (Key Biodiversity Areas [panel **b**] and Remaining Wilderness [panel **c**]). Mining areas were mapped using a 50-cell radius around known mining properties, separated into those that are listed as being in pre-operational, operational, and closed phases. Stars denote significance differences (*$p < 0.05$, ***$p < 0.0001$) among areas targeting different commodity types (critical, other, and both) within a mining phase (pre-operational, operational, and closed). Results hold when defining mining areas using a 10-cell radius around mining properties and when sampling at 300 km intervals. *Note*: Graphed proportions were calculated from full extent of mining and non-mining areas, whereas reported statistics were calculated using the 1 km$^2$ cells sampled at 100-cell intervals.

overlap for areas with operational (7.5% vs. 7.1%; chi-squared = 0.03, $p = 0.86$) and closed mining properties (8.0% vs. 4.4%; chi-squared = 2.13, $p = 0.14$), but significantly less overlap for areas with pre-operational properties (13.9% vs. 21.6%; chi-squared = 22.57, $p < 0.001$; Supplementary Data 1). The large extent of pre-operational mines targeting coal and other non-critical materials in artic and sub-arctic North America and Asia (Fig. 1) likely explain these results; production faces large economic hurdles in these regions. However, similar to PAs, critical mining areas overlapping Remaining Wilderness were significantly more dense than other mining areas overlapping Remaining Wilderness (3.1 vs. 1.9; $D = 0.15$, $p = 0.01$; Supplementary Data 3). These threats are concerning, given that wilderness lacks formal protection[36] and any new infrastructure will erode its ecological integrity[27].

**Future mining threats to biodiversity**. The global area influenced by mining will almost certainly grow in extent and density in future, and the increased demand for renewable energy technologies and infrastructure will likely be one contributing factor. While diverting some of the materials used in non-renewable energy infrastructure may minimize threats of renewable energy production to biodiversity, fossil fuels will still likely play an important role in meeting the future energy demands of a growing global population. We discovered a greater proportion of pre-operational mines targeting materials needed for renewable energy production (83.9%) compared to operational mines targeting these materials (72.8%; Supplementary Table 2), and that pre-operational mining areas targeting the materials critical for renewables also seem more dense than those targeting other materials (3.2 vs. 2.7; Supplementary Table 2). Increasing the extent and density of mining areas will obviously cause additional threats to biodiversity, and our analysis reveals that a greater proportion of mines targeting materials for renewable energy production may further exacerbate threats to biodiversity in some areas (here demonstrated by their increased mining density within Key Biodiversity Areas and Remaining Wilderness at the global scale).

Careful strategic planning is urgently required to ensure that mining threats to biodiversity caused by renewable energy production do not surpass the threats averted by climate change mitigation and any effort to slow fossil fuel extraction and use. Habitat loss and degradation currently threaten >80% of endangered species, while climate change directly affects 20%[37]. While we cannot yet quantify potential habitat losses associated with future mining for renewable energies (and compare this to any reduced risks of averting climate change), our results

illustrate that associated habitat loss could be a major issue. At the local scale, minimizing these impacts will require effective environmental impact assessments and management. Importantly, all new projects must adhere strictly to the principals of the Mitigation Hierarchy[38], where biodiversity impacts are first avoided where possible before allowing compensation activities elsewhere. While compensation may help to overcome some of the expected biodiversity impacts of mining in some places[39], rarely does this approach achieve No Net Loss outcomes universally[39,40].

There is urgent need to understand the size of mining risks to biodiversity (climate change, and efforts to avert it) and strategically account for them in conservation plans and policies. Yet, none of these potential tradeoffs are seriously considered in international climate policies[3], nor are new mining threats addressed in global discussions around post-2020 United Nation's Strategic Plan for Biodiversity[28,41]. Necessary actions include strengthening policies to avoid negative consequences of mining in places fundamentally important for conservation outcomes, and developing necessary landscape plans that explicitly address current and future mining threats. These actions must also be supported by a significant research effort to overcome current knowledge deficits. A systematic understanding of the spatially explicit consequences (rather than potential threats, as investigated here) of various mining activities on specific biodiversity features, including those that occur in marine systems and at varying distances from mine sites (rather than within a predefined distance of 50 km, as done here), is required.

## Methods
**Minerals critical for renewable energies**. We identified the materials required for renewable energy production (Supplementary Table 1), using projected material demand under 2050 low-carbon energy scenarios[6]. These projections focused on two technologies (wind turbine manufacturing and solar photovoltaic installations; both expected to experience huge growth in future) and one energy-using technology (storage batteries for electric vehicles, which will address transportation emissions). However, we also included the minerals required for other technologies[6] (including carbon capture and storage installations, nuclear electricity generation installations, LED manufacturing, electric vehicle manufacturing, and lithium-Ion batteries) to get a better split between renewable and non-renewable energies.

**Mapping mining areas**. We obtained point locations for mining properties worldwide[31], including pre-operational, operational, and closed sites (Supplementary Table 3). We used Mollweide equal area projection to analyze all data in ArcGIS 10.6. We mapped the global extent of areas potentially influenced by mining by summing the number of mining properties within a 50-cell radius of each 1 km$^2$ grid cell containing a mining property and clipping resultant areas to the terrestrial land surface, to include both the direct and indirect impacts of

mining on biodiversity. Counts represent mining density, and we converted density maps to binary values to indicate mining influenced areas when mining density was >0.

We chose a 50-cell radius (~50 km), as mining threats are found to extend this distance from mining properties[20]. However, given that the magnitude of direct and indirect effects likely varies tremendously among mines[42] (e.g. extracting different minerals, with different methods), we also analyzed a more conservative radius of 10 km. This analysis shows that, although the extent of mining-influenced areas naturally decreases (49 million km$^2$ using a 50-cell radius vs. 6.6 million km$^2$ using a 10-cell radius; Supplementary Fig. 1; Supplementary Table 1) and their proportional overlap with conservation priorities changes slightly (Supplementary Figs. 2 and 3), most relative differences between mining and non-mining areas and between critical and other mining areas did not (Supplementary Data 1).

We repeated this process for subsets of mining areas by development status (pre-operational, operational, closed; Supplementary Table 3) and commodity type (critical areas only, other areas only, and areas targeting both commodity types; Supplementary Table 1). Note other mining areas included coal mines, some of which produce metallurgic coal used in steel production, which in 2018 represented 12% of world coal production[43], and thus may also be influenced by increasing demand for renewable energy infrastructure. We mapped non-mining areas as any land (excluding Antarctica) outside mining influenced areas.

We did not examine the changes in mineral demand specifically driven by renewable energy production. These dynamics are highly uncertain and dependent on many factors, most notably including the mix of technologies, infrastructure, and the strategies ultimately used to mitigate carbon emissions and delivery of future energy demands. Instead, out method examines whether the minerals critical to produce these technologies and infrastructure were present or absent. Our critical mining areas should be considered as the sites potentially influenced by increasing demand for renewable energy technologies and infrastructure, rather than illustrative of the relative distribution of threats, which would be influenced by demand for other unrelated products (e.g. steel). Further analyses, which were also beyond our scope, could examine the distribution of specific materials required for certain products or technologies.

**Biodiversity conservation sites**. We obtained spatial data on current PAs[24] and other conservation priorities (Key Biodiversity Areas[21] and Remaining Wilderness[25]) across terrestrial systems. We analyzed 28,409 PAs (23.38 million km$^2$) formally designated for conserving biodiversity and ecosystem services[44]. PAs cover various governance types (public, private, and indigenous/community areas) and management categories (from strict nature reserves to those permitting managed extraction) and are central to the Convention on Biological Diversity's Strategic Plan and meeting 2020 Aichi Targets[45]. For conservation priorities, we analyzed 13,320 Key Biodiversity Areas (13.87 million km$^2$), representing sites that contribute significantly to global biodiversity persistence. Sites qualify for designation when they meet one of 11 criteria across five categories (threatened biodiversity, geographically restricted biodiversity, ecological integrity, biological processes, irreplaceability)[21]. We also analyzed Earth's Remaining Wilderness—areas free from the industrial-scale activities and human pressures that cause significant biophysical disturbance. Specifically, we used the 2009 Last of the Wild indicator[25], which identifies the top 10% of intact habitats for each of Earth's 60 biogeographic realms (12.12 million km$^2$). Wilderness areas are not formally considered in global conservation policies[36], but are attracting attention as a proactive means to protect biodiversity[46]. Some areas designated as Remaining Wilderness overlap with PAs (2.47 million km$^2$) and Key Biodiversity Areas (2.28 million km$^2$).

**Data analysis**. We sampled mining and non-mining areas at 100-cell (~100 km) intervals ($n = 13,483$), to remove potential issues with large sample sizes and the spatial autocorrelation created in our dataset by mapping mining areas using a 50-cell radius around each mining property. Sampling at 100-cell intervals also removed most other sources of spatial autocorrelation (Supplementary Table 4). We used GeoDa to produce correlograms[47] quantifying correlation in mining density within 50 km distance bands (with a random sample of 50,000 cells and 1 million cell pairs). For mining areas created using a 50-cell radius, correlation by 100 km was 0.002 for areas targeting other minerals, 0.07 for areas targeting critical minerals, and 0.1 for both. For mining areas created using 10-cell radius, correlation at 100 km was 0.003 for others, 0.009 for critical, and 0 for both.

We overlaid the sampled data with PAs, Key Biodiversity Areas and Remaining Wilderness to determine differences in their proportional overlap (two-sided Chi-squared tests) and average mine density (Kolmogorov–Smirnov tests) for (1) mining vs. non-mining areas; (2) critical vs. other mining areas, and (3) pre-operational vs. operational vs. closed mining areas.

Finally, to test the robustness of our results to the 100-cell sampling interval, we sampled our datasets at 300-cell intervals (starting at a different cell to ensure an entirely different set of sample cells) and repeated all analyses. We found that the relative differences between mining and non-mining areas and between critical and other mining areas did not change, although the significance of these differences sometimes did (Supplementary Data 1–3).

**Reporting summary**. Further information on research design is available in the Nature Research Reporting Summary linked to this article.

## Data availability
Spatial data on Protected Areas and Conservation Priorities are available from refs. [21,24,25]. The mining properties database[31] is not freely available, but global mining areas maps produced in this study (1 km$^2$ resolution, using 50-cell radius, per commodity type (critical, other and both; Fig. 1) can be downloaded as tiff files from https://doi.org/10.6084/m9.figshare.12630092. All other combinations of mining areas (i.e. by status [pre-operational, operational, and closed] and when using 10-cell radius; Supplementary Fig. 1) can be obtained by contacting the corresponding author.

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

## Acknowledgements
The authors thank K. Jones and J. Allen for providing data, S. Blomberg and J. Rhodes for providing statistical advice, and H.P. Possingham, T.H. Ricketts, M. Maron, and S.H. Ali for helpful discussions and feedback on a previous version of this manuscript. L.J.S. acknowledges an Australian Research Council Discovery Early Career Researcher Award (DE170100684) and M.C.D. was supported by the Sustainable Minerals Institute's Complex Orebodies program.

## Author contributions
L.J.S. conceived the idea and designed the research; L.J.S. and M.C.D. collated datasets and performed analysis; L.J.S., M.C.D., J.E.M.W., and R.K.V. interpreted results and wrote the manuscript.

## Competing interests
The authors declare no competing interests.
