## [Peer Review File · Nature Communications]

Reviewers' comments:

Reviewer #1 (Remarks to the Author):

Thank you for your submission to Nature Communications on the biodiversity threats arising from mining of materials required for the foreseen growth in renewable energy.

Overall the manuscript is logically structured and comes to an interesting conclusion. I would hope to see the following addressed before publication:

What is the overlap between KBAs and LOWs are they mutually exclusive, one a subset of the other, some overlap?

What does 'influenced' mean 37% Mining influenced (line 58)– I'm most familiar with Australia as a mining nation but my recollection was that the affected land footprint of mining was below 2%, thus for me the figure of 37% seems high.

If you're saying 'influenced' is 'within 50km of mine site' then say this explicitly in the 'mapping mining areas' section. I wonder if 'potential influence' is a more helpful term.

Lines 134-136 present an important (and big) statement – is there any way to ground this in relative (even approximate) quantification of the biodiversity threat caused by mining and that averted by climate change mitigation, otherwise the assertion may come across as inflated.

Fig 1, colour scheme may be unhelpful for RGB colour-blind people, I'd also prefer 'both types' as red

Looking at Table S1 - how much of final demand for the listed elements is linked with renewables? Whilst this data may be difficult to assemble, perhaps at least some ranges – otherwise a metal which may have a small percentage of its final demand going to renewables yet has the total impact of all production attributed to it (if I understand correctly). For me, this requires attention /clarification.

Reviewer #2 (Remarks to the Author):

This is an interesting and timely paper addressing the issues of the local biodiversity impacts of scaling up mining to produce resources needed to mitigate climate change. I was pleased to review this and I think it contains valuable information. I have three concerns that I think if addressed could improve the contribution that the manuscript could make.

First, I think the characterisation of minerals as "renewable" or "non-renewable" based on whether the mineral is used in the development of renewable technologies seems simplistic. The authors make no attempt to explore the relative importance of the demand from renewable technologies for these minerals and therefore the extent to which scaled up demand from the renewables sector would drive increases in mining. Fundamentally, it seems misleading to imply that renewable technologies are underpinning the threat to biodiversity from mining for minerals that amongst all their other uses are also used in renewable technologies. Without this exploration and even a first order justification I find the conclusions of the paper might not be valid. I think this issue comes to a head with the statement on L124 "The global area influenced by mining will almost certainly grow in extent and density to support future renewable energy production". This is at present an assumption and represents more of a scenario than reality without further analysis and justification.

Second, I think the methodological section of the manuscript is too brief to the extent that it is

hard to evaluate the manuscript. The methodological description makes very little attempt to ensure that the analyses described can be reproduced. I think the manuscript would be greatly improved if the methods were more comprehensive and future researchers would benefit from analysis code being openly available alongside data.

Methodological concerns to be addressed include:

1. Better explanation of 50km buffer (is this confounding direct and indirect impacts). How sensitive are the findings to this buffer size, it seems entirely arbitrary at present, so the reader needs to know what the implications are of using this value.
2. Linked to 1 above, how is the 50km buffer used in the sampling of overlaps?
3. Sampling and statistical analysis needs better explanation – what are the implications of throwing away data from the analysis and how did you ensure your approach removed autocorrelation problems?
4. The chi-square test details are very limited. What is the test being performed – the manuscript implies this is a test where the null hypothesis is that there is no difference between the observed counts and expected counts assuming that mining and non-mining are equally common – if so I don't think this is the correct test. Instead it should be testing if the observed differs from the expected proportions given that we know a priori that there is a greater area of non-mining than mining.

Third, I think that the discussion really needs to be expanded in two directions. Firstly, how can renewable technologies be expanded without expanding mining (e.g. shifting mineral uses between non-renewable technologies and renewable technologies) or the impacts from mining (by mitigating impacts)? Second, how important might these local biodiversity impacts be relative to the potential impacts from climate change?

I also hope that addressing the following issues would improve the balance of the manuscript.

L28: The USGS, 2019 mineral estimates for Lithium state that Argentina possesses the largest Lithium reserves (14.5 Mt), whilst Bolivia, Chile and Australia have 9, 8.5 and 7.7 Mt respectively. I feel that the example is selective and possibly misleading – for example are Bolivia's Lithium reserves located underneath or proximal to important biodiversity areas in Bolivia?

L67: "As a result, production costs per mine for coal are more dependent on pre-existing proximate processing and transportation infrastructure, leading to a closer clustering of these operations." This is a strong assertion is this actually true? It would be stronger if there was some support.

L77-79: "Alternatively, it is possible that PAs have been sited away from mineral resources; however, this is less likely to explain differences between mining and non-mining areas, given that overlap with pre-operational mining areas" – a more comprehensive discussion here is needed. The options seem to be (though this may not be exhaustive): PAs are not located in areas thought to overlay minerals; mining companies do not explore for minerals near PAs; mining companies do not exploit resources underneath PAs even if they do identify resources there; PAs are downsized to remove overlaps with mineral resources. The supplementary table S4 (which is never referenced in the main text!) could be useful as well – it shows that there is lower mining density in conservation areas that outside, possibly indicating an effect. However, I feel that the 50km aggregation could be introducing spatially inaccurate results here though the methods are not sufficient to identify if there might be issues.

L88-90: "While mineral prospecting does not (yet) cause the same threats to biodiversity as operational mining sites do, they are not without risk". This deserves greater discussion, there is considerable evidence that exploration can have a substantial impact (see e.g. citations in Parish et al., 2013: DOI 10.1007/s00267-012-9983-6, about environmental impacts of seismic surveys). In particular, if exploration is damaging, then much of the impacts assigned to the future mining activities might already have been realised.

Reviewer #3 (Remarks to the Author):

Renewable energy production will exacerbate mining threats to biodiversity

Remarks to the Author

The authors set out to map the global distribution of mining areas and how it overlaps with protected areas and other places of conservation priority. Most of the world's mining areas target materials needed for renewable energy production and, as the demand for these resources grows, mining will increasingly encroach on biodiverse lands. It is suggested that "mining threats to biodiversity may surpass those averted by climate change mitigation" through renewable energy production. The study, therefore, flags the need for strategic planning to mitigate this.

With an impressive data set, they used a geospatial approach to compare point locations of mining sites with Protected Areas, Key Biodiversity Areas and Last of the Wild lands. They developed an index of mine density by quantifying the number of mining properties within a 50 km sphere of influence from every cell in a 1 km raster. Pre-operational, operational and closed status of mines was considered, and so were areas where no mining occurred. Importantly, the mines were categorized as targeting materials for renewable energy production ("renewable areas") and targeting other materials ("non-renewable areas"). It is a competent data analysis that supports the thesis outlined above.

Having said that, the mapping approach would be more nuanced if different types of mines (i.e., targeted materials and associated mining methods) could be differentiated in terms of their area of impact. Different mining activities in different environments do not have a constant surrounding sphere of impact on biodiversity. A 50 km (or greater) zone might be a reasonable assumption in some environments and for some mining activities, but readers need to be convinced that applying a standard 50 km footprint around all ~62,000 mining points in this global data set is justifiable. Another concern is that coal is categorized as a non-renewable mining target (Table S1). But a portion of the world's mined coal, somewhere between 10% and 15%, is targeted for metallurgy, which then supports renewable energy infrastructure. Acknowledging this within the manuscript would strengthen the authors thesis.

Response to reviewers' comments:

Reviewer #1 (Remarks to the Author):

1. Thank you for your submission to Nature Communications on the biodiversity threats arising from mining of materials required for the foreseen growth in renewable energy. Overall the manuscript is logically structured and comes to an interesting conclusion. I would hope to see the following addressed before publication:

RESPONSE: We thank the reviewer for their helpful suggestions. We have addressed each of them in detail below. Please note that line and page numbers quoted in our response refer to our revised marked-up manuscript (pdf version, showing in-line edits).

2. What is the overlap between KBAs and LOWs are they mutually exclusive, one a subset of the other, some overlap?

RESPONSE: There is some overlap between LoW and KBAs and PAs. We have now reported this in our methods: “*Some areas designated as LoW overlap with PAs (2.47 million km²) and KBAs (2.28 million km²)*” (P5 L148–249).

3. What does ‘influenced’ mean 37% Mining influenced (line 58) – I’m most familiar with Australia as a mining nation but my recollection was that the affected land footprint of mining was below 2%, thus for me the figure of 37% seems high.

RESPONSE: These results refer to the overlap between ‘mining influenced areas’ (mapped using a 50 km buffer around each mine site) and our indicators of conservation priorities. We have now clarified this term in our Methods (P4 L199), specifically stating that these influences include the direct impacts of mining (e.g. vegetation clearing for mineral extraction) and indirect off-site impacts (P4 L201–202). We have also explained this method earlier in the manuscript and added that this 37% figure assumes that indirect impacts extent 50km from mining sites (P1 L50–P2 L52).

4. If you’re saying ‘influenced’ is ‘within 50km of mine site’ then say this explicitly in the ‘mapping mining areas’ section. I wonder if ‘potential influence’ is a more helpful term.

RESPONSE: We have clarified our definition of mining-influenced areas in our manuscript and explained that this includes both the direct and indirect threats of mining to biodiversity (P1 L50–P2 L52). We also now use the term “potentially influence” throughout the manuscript (P1 L8; P1 L48; P2 L67; P2 L82; P4 L199; P5 L227).

5. Lines 134-136 present an important (and big) statement – is there any way to ground this in relative (even approximate) quantification of the biodiversity threat caused by mining and that averted by climate change mitigation, otherwise the assertion may come across as inflated.

RESPONSE: We agree that this is a major statement. Habitat loss and degradation currently threatens >80% of endangered species, while climate change directly affects 20% (see Maxwell et al. 2016, Nature; doi:10.1038/536143a; P4 L159–160). While we cannot yet quantify potential habitat losses associated with future mining for renewable energies (and compare this to the reduced risks of averting climate change that may occur with this

transition to renewable energy), our results illustrate that associated habitat loss is likely a major issue for many species and important biodiversity sites. We have addressed this comment by reiterating the urgent need to understand the size of these risks (climate change, and efforts to avert it), and strategically account for these risks in biodiversity plans and policies (P4 L170–171).

6. Fig 1, colour scheme may be unhelpful for RGB colour-blind people, I'd also prefer 'both types' as red

RESPONSE: We thank the reviewer – we have edited the colour scheme used in Figure 1.

7. Looking at Table S1 - how much of final demand for the listed elements is linked with renewables? Whist this data may be difficult to assemble, perhaps at least some ranges – otherwise a metal which may have a small percentage of its final demand going to renewables yet has the total impact of all production attributed to it (if I understand correctly). For me, this requires attention /clarification.

RESPONSE: We thank the reviewer for this comment. We did not examine the changes in mineral demand specifically driven by renewable energies. These dynamics are highly uncertain and dependent on factors related to the mix of technologies, infrastructure and strategies used to mitigate carbon emissions. Instead, we mapped whether the minerals critical to produce these technologies and infrastructure were present or absent. We agree that understanding changes in relative demand is necessary to identify the places most at risk *within* our mapped areas; however, this is well beyond the scope of this study. We have now added this clarification to our Methods (P5 L222–226) and edited our manuscript to clarify that our maps do not illustrate the relative distribution of threats, which will also be influenced by demand for other unrelated products (e.g. steel), but simply illustrate where these threats occur (P5 L226–230).

Reviewer #2 (Remarks to the Author):

8. This is an interesting and timely paper addressing the issues of the local biodiversity impacts of scaling up mining to produce resources needed to mitigate climate change. I was pleased to review this and I think it contains valuable information. I have three concerns that I think if addressed could improve the contribution that the manuscript could make.

RESPONSE: We thank the reviewer for their helpful suggestions. We have addressed each of them in detail below. Please note that line and page numbers quoted in our response refer to our revised marked-up manuscript (pdf version, showing in-line edits).

9. First, I think the characterisation of minerals as “renewable” or “non-renewable” based on whether the mineral is used in the development of renewable technologies seems simplistic. The authors make no attempt to explore the relative importance of the demand from renewable technologies for these minerals and therefore the extent to which scaled up demand from the renewables sector would drive increases in mining. Fundamentally, it seems misleading to imply that renewable technologies are underpinning the threat to biodiversity from mining for minerals that amongst all their other uses are also used in renewable technologies. Without this exploration and even a first order justification I find the conclusions of the paper might not be valid. I think this issue comes to a head with the

statement on L124 “The global area influenced by mining will almost certainly grow in extent and density to support future renewable energy production”. This is at present an assumption and represents more of a scenario than reality without further analysis and justification.

RESPONSE: Agreed. We no longer characterise our mining influenced areas as renewable vs. non-renewable. Instead, we refer to these places as areas containing minerals critical for renewable energy infrastructure and technologies – “critical mining areas” vs. “other mining areas”.

We note that we did not examine the changes in mineral demand specifically driven by renewable energies. These dynamics are uncertain and dependent on factors such as the mix of technologies and strategies used to mitigate emissions. Instead, we mapped whether the minerals critical to produce these technologies and infrastructure were present or absent within mining influenced areas. We agree that understanding the changes in relative demand is necessary to identify the places most at risk *within* our mapped areas; however, this was beyond our study scope. We have added this clarification to our Methods (P5 L222–226) and edited our manuscript to clarify that our maps do not illustrate the relative distribution of threats, which will also be influenced by demand for other unrelated products (e.g. steel), but simply illustrate where these threats occur (P5 L226–230).

Specifically in response to the statement cited on L124, we have clarified that the global area influenced by mining will grow and that increasing demand for renewable energy production will be one contributing factor driving consequent threats to biodiversity (P3 L143–145).

10. Second, I think the methodological section of the manuscript is too brief to the extent that it is hard to evaluate the manuscript. The methodological description makes very little attempt to ensure that the analyses described can be reproduced. I think the manuscript would be greatly improved if the methods were more comprehensive and future researchers would benefit from analysis code being openly available alongside data.

RESPONSE: Agreed. We have now expanded our Methods section to explain comprehensively our analyses to ensure reproducibility. If accepted for publication, we will make our datasets and modelled results publically available, as per the journal requirements.

Methodological concerns to be addressed include:

11. 1. Better explanation of 50km buffer (is this confounding direct and indirect impacts). How sensitive are the findings to this buffer size, it seems entirely arbitrary at present, so the reader needs to know what the implications are of using this value.

RESPONSE: We used a 50 km radius to capture the direct and indirect effects of mining on biodiversity (P1 L50–51; P5 L210–211). The 50km radius was chosen based on previous studies; however, given that the magnitude of indirect effects will likely vary tremendously among commodities and extraction techniques, we have now also included a sensitivity analysis using an alternative (and more conservative) radius of 10 km (P1 L51–52; P5 L216–218).

As now explained in the manuscript, all reported results held true when using a 10 km radius, except two. (1) The total area influenced by mining, which is 6.6 million km² if using a 10km

radius (P5 L218). (2) The significant difference reported for critical mining areas overlapping PAs being denser than other mining areas overlapping PAs (P3 L128–129).

All Figures and Tables in the main text and Supplementary Information now show the results for both a 50 km and 10 km radius (Fig S1–S3; Tables S2–S5).

12. 2. Linked to 1 above, how is the 50km buffer used in the sampling of overlaps?

RESPONSE: As we now clarified, the 50 km buffer was used to define our mining areas to capture direct and indirect threats of mining to biodiversity (P1 L50–51; P5 L210–211). Sampling was done independent to this buffer.

13. 3. Sampling and statistical analysis needs better explanation – what are the implications of throwing away data from the analysis and how did you ensure your approach removed autocorrelation problems?

RESPONSE: We have now expanded our Methods and we thank the reviewer for these suggestions. We subsampled at 100 km intervals to remove the spatial autocorrelation created by mapping mining areas using a 50 km buffer width (P5 L264–265). We did not quantify or remove other potential sources of autocorrelation (P6 L266).

14. 4. The chi-square test details are very limited. What is the test being performed – the manuscript implies this is a test where the null hypothesis is that there is no difference between the observed counts and expected counts assuming that mining and non-mining are equally common – if so I don't think this is the correct test. Instead it should be testing if the observed differs from the expected proportions given that we know a priori that there is a greater area of non-mining than mining.

RESPONSE: We have now clarified that the Chi-squared tests compared proportions: e.g. the proportion of mining influenced areas that overlapped with conservation priorities vs. the proportion of non-mining areas that overlapped with conservation priorities (P6 L266).

15. Third, I think that the discussion really needs to be expanded in two directions. Firstly, how can renewable technologies be expanded without expanding mining (e.g. shifting mineral uses between non-renewable technologies and renewable technologies) or the impacts from mining (by mitigating impacts)? Second, how important might these local biodiversity impacts be relative to the potential impacts from climate change?

RESPONSE: We thank the reviewer for these suggestions. We have added text and references to the Introduction to suggest that expanding renewable energy production is unlikely achievable without expanding mining (see Ali et al. 2017 *Nature* 543 367–372; P1 L28–29) but, in the Discussion, also suggest that impacts on biodiversity may be minimised by diverting materials away from non-renewable technologies where possible (P3 L154–157).

We have now discussed the importance of ensuring new mining projects adhere to the Mitigation Hierarchy – where impacts are first avoided, before minimising and compensating impacts by undertaking conservation activities elsewhere (P4 L173–175). While compensation may help to mitigate some biodiversity impacts of mining, rarely do they achieve No Net Loss outcomes (P4 L175–177).

We have also added more detail in regards to the relative importance of local biodiversity impacts compared to those from climate change (P4 L168–172). Habitat loss and degradation currently affects around 80% of endangered species, while climate change impacts 20% (see Maxwell et al. 2016; Nature; doi:10.1038/536143a). Any future habitat loss from mining is likely to be large, given that this is the biggest threat to biodiversity already. See also our response to Reviewer 1 comment #5.

16. I also hope that addressing the following issues would improve the balance of the manuscript. L28: The USGS, 2019 mineral estimates for Lithium state that Argentina possesses the largest Lithium reserves (14.5 Mt), whilst Bolivia, Chile and Australia have 9, 8.5 and 7.7 Mt respectively. I feel that the example is selective and possibly misleading – for example are Bolivia's Lithium reserves located underneath or proximal to important biodiversity areas in Bolivia?

RESPONSE: Reviewer 2 is correct that countries well suited to supply lithium to meet future renewable energy demand include Argentina, Bolivia, Chile and Australia. We highlight the case of Bolivia for three reasons (now explained on P1 L30–33). First, Bolivia has the second largest untouched lithium reserve in the world. Second, this lithium reserve co-occurs with the Salar de Uyuni salt pan (a fragile, yet biodiverse ecosystem). Third, mining in this region is scarce, meaning that new mines may have larger impacts on untouched biodiversity than those in other already developed regions.

17. L67: “As a result, production costs per mine for coal are more dependent on pre-existing proximate processing and transportation infrastructure, leading to a closer clustering of these operations.” This is a strong assertion is this actually true? It would be stronger if there was some support.

RESPONSE: We have provided further clarification on this statement (P2 L75–83).

“For example, annual coal production (7,813 Mt in 2018)²⁵ greatly exceeds that of copper metal (21 Mt)²⁶ (i.e. a commodity required for renewable energies; Table 1). Most copper is produced in the form of concentrate, so the tonnage of material transported in a similar way to coal would be in the order of 80Mt. Thus, the world coal industry transports two orders of magnitude (100x) more material than the copper industry. This is supported by production data per mine²⁷, showing that the average material transported from a coal mine is more than 10 times the amount of concentrate shipped from a copper mine. As a result, costs per mine for coal are more dependent on pre-existing proximate processing and transportation infrastructure, leading to a closer clustering of these operations.”

18. L77-79: “Alternatively, it is possible that PAs have been sited away from mineral resources; however, this is less likely to explain differences between mining and non-mining areas, given that overlap with pre-operational mining areas” – a more comprehensive discussion here is needed. The options seem to be (though this may not be exhaustive): PAs are not located in areas thought to overlay minerals; mining companies do not explore for minerals near PAs; mining companies do not exploit resources underneath PAs even if they do identify resources there; PAs are downsized to remove overlaps with mineral resources. The supplementary table S4 (which is never referenced in the main text!) could be useful as well – it shows that there is lower mining density in conservation areas that outside, possibly indicating an effect. However, I feel that the

50km aggregation could be introducing spatially inaccurate results here though the methods are not sufficient to identify if there might be issues.

RESPONSE: We thank the reviewer for this suggestion. We have now rewritten this paragraph for clarify and have included reference to the results contained in Table S4 (P2 L91–107):

“Approximately 8% of the global area potentially influenced by mining overlapped with PAs (Fig. 2), indicating extensive threats to currently protected biodiversity. However, these threats were significantly less than expected when compared to the overlap between non-mining areas and PAs (8.4% vs. 16.7%; $\chi^2=28.13$, $p<0.001$; Table S3), possibly suggesting PAs do prevent some mining activities within their boundaries. It is also possible that mineral exploration occurs less often within PAs and this, in turn, lowers the rate of mining in these areas. We found less overlap between pre-operational mining areas (where minerals are not yet confirmed economically viable) and PAs than non-mining areas and PAs (12.2%; $\chi^2=55.54$, $p<0.001$; Table S3), although pre-operational mining density did not differ significantly between areas overlapping PAs and areas not overlapping PAs (4.0 vs. 4.4 mining properties; Table S4). Alternatively, it is possible that PAs are sited away from mineral resources or that, discovering minerals within PAs, has led to their abolishment. Since evidence does exist for mineral resources leading to world-wide downgrading, downsizing and degazettement of PAs (PADDD)²⁸, further reducing mining threats to PAs, by strengthening their governance and management, is crucial if conservation efforts are to secure their biodiversity²⁹.”

19. L88-90: “While mineral prospecting does not (yet) cause the same threats to biodiversity as operational mining sites do, they are not without risk”. This deserves greater discussion, there is considerable evidence that exploration can have a substantial impact (see e.g. citations in Parish et al., 2013:DOI 10.1007/s00267-012-9983-6, about environmental impacts of seismic surveys). In particular, if exploration is damaging, then much of the impacts assigned to the future mining activities might already have been realised.

RESPONSE: We agree that mineral exploration can negatively affect biodiversity and have added the suggested reference to our study (P3 L114–115). We disagree that exploration impacts may render future extraction impacts insignificant. While exploration may lead to the introduction of invasive species or loss of habitat for seismic surveys and drilling, extraction will almost always lead to additional habitat clearing and ecological damage.

Reviewer #3 (Remarks to the Author):

20. The authors set out to map the global distribution of mining areas and how it overlaps with protected areas and other places of conservation priority. Most of the world’s mining areas target materials needed for renewable energy production and, as the demand for these resources grows, mining will increasingly encroach on biodiverse lands. It is suggested that “mining threats to biodiversity may surpass those averted by climate change mitigation” through renewable energy production. The study, therefore, flags the need for strategic planning to mitigate this.

With an impressive data set, they used a geospatial approach to compare point locations of mining sites with Protected Areas, Key Biodiversity Areas and Last of the Wild lands. They developed an index of mine density by quantifying the number of mining properties within a 50 km sphere of influence from every cell in a 1 km raster. Pre-operational,

operational and closed status of mines was considered, and so were areas where no mining occurred. Importantly, the mines were categorized as targeting materials for renewable energy production (“renewable areas”) and targeting other materials (“non-renewable areas”). It is a competent data analysis that supports the thesis outlined above.

RESPONSE: We thank the reviewer for their helpful suggestions. We have addressed both of them in detail below. Please note that line and page numbers quoted in our response refer to our revised marked-up manuscript (pdf version, showing in-line edits).

21. Having said that, the mapping approach would be more nuanced if different types of mines (i.e., targeted materials and associated mining methods) could be differentiated in terms of their area of impact. Different mining activities in different environments do not have a constant surrounding sphere of impact on biodiversity. A 50 km (or greater) zone might be a reasonable assumption in some environments and for some mining activities, but readers need to be convinced that applying a standard 50 km footprint around all ~62,000 mining points in this global data set is justifiable.

RESPONSE: We used a 50 km buffer to capture the direct and indirect impacts of mines on biodiversity (now clarified on P1 L50–51; P5 L210–211). The 50 km distance was based on previous analyses involving large metal mines (see Sonter et al. 2017 *Nature Communications*, doi: 10.1038/s41467-017-00557-w). We agree indirect effects would vary tremendously among mines targeting different commodities and using different extraction techniques, but, although quantifying these effects would be interesting, it is beyond our study scope. To address this comment, as well as those made by Reviewer 1 and 2 (comments #4 and #11, respectively), we have now included an analysis using a more conservative radius of 10 km. All reported results hold when using 10 km radius, except two: (1) the total area influenced by mining, which is 6.6 million km² if using a 10 km radius (P5 L218) and (2) the significant difference reported for critical mining areas overlapping PAs being more dense than other mining areas overlapping PAs (P3 L128–129). All Figures and Tables in the main text and Supplementary Information now show the results for both a 50 km and 10 km radius (See Fig S1–S3; Tables S2–S5).

22. Another concern is that coal is categorized as a non-renewable mining target (Table S1). But a portion of the world’s mined coal, somewhere between 10% and 15%, is targeted for metallurgy, which then supports renewable energy infrastructure. Acknowledging this within the manuscript would strengthen the authors thesis.

RESPONSE: We have clarified this limitation in our manuscript (P5 L225–228) to suggest that some of regions that produce metallurgic coal may indeed also be influenced by increasing demand for renewable energies. Specifically, the International Energy Agency (2019) reported 2018 production of 5967.8Mt of Steam coal, 1033.3Mt of coking coal, and 803.2Mt of lignite. Coking coal therefore represents about 12% of world coal production.

Reviewers' comments:

Reviewer #1 (Remarks to the Author):

Thank you for an updated manuscript which has addressed the key points needing attention.

Reviewer #2 (Remarks to the Author):

The authors have responded thoroughly to the comments provided in the first review and this has, in my opinion, improved the article. I agree with the central message that we need to plan future mining developments carefully during the transition to a low carbon economy and mitigate impacts on biodiversity as far as possible. However, I still have a couple of comments that I would like to see addressed that I hope increase the robustness of the findings. Most of these relate to the message that critical mines might have a disproportionately greater impact than other mining.

L132 – 134: “but increasing the proportion of mines targeting materials critical for renewable energy production may also disproportionality increase the threats to biodiversity within currently protected areas.” I think this message probably needs to be expanded reflecting on the types of mines and how their impacts are managed given that the disproportionate impact is only based on the density of critical mining operations, a finding that is not robust to the impact radius of each mine. And a result that hasn't been tested against some of the methodological assumptions – see below.

I am still concerned about two aspects of the analysis particularly given the slightly nuanced messages when comparing critical and other mining sites.

The first regards the sampling approach. The authors give no justification for the 100km cell interval, which is entirely arbitrary. Given that most of the findings comparing “critical” and “other mining” areas depend on this sampling, to what extent are the results dependent on this interval? To ensure that the results are robust, I would recommend that the authors do two things. First, justify why the sampling approach is needed, is this primarily to reduce data volume to make the analysis tractable? If it is removing autocorrelation because that breaks the assumption of data independence in the statistical tests, then the authors should show that the autocorrelation is removed at this scale or use a test that can account for spatial autocorrelation. Second, would a different analysis distance or does the exact 100km grid impacts on the findings? The authors could test this by sampling at 100km (and another valid distance) using a set of different origins to generate a set of different samples and test that their findings are robust across these.

The second regards the buffering. Given the findings do show some sensitivity to the impact radius, does that not raise the question of how severe impacts are spread across the distance from the mine, in other words an impact decay kernel around each mine. I am not asking the authors to repeat their analysis with this approach as that would be a significant amount of work and the exact nature of such a kernel is uncertain, however, I think the manuscript would be improved if the discussion acknowledged some of the possible methodological issues associated with their analysis and possible avenues to improve this.

L52: “We distinguish areas targeting the materials critical for renewable energy production” could possibly more reasonably be reworded as “We distinguish areas targeting the materials that amongst many other uses in the present day are also critical for renewable energy production” or words to that effect.

L211: “Counts represent mining density, and we converted density maps to binary values to indicate mining influenced areas.” For clarity it would be important to state what is left to be assumed here that this was equal to 1 where the density was greater than 0, and 0 when the density was equal to 0.

Reviewer #3 (Remarks to the Author):

I had only a few noteworthy concerns about the original manuscript, which have now been addressed to my satisfaction in the revised manuscript.

I suggested that the mapping approach would be more nuanced if different types of mines (in terms of targeted materials and associated mining methods) could be differentiated by their respective areas of impact. The authors have argued that this is beyond the scope of their work, and I do accept this limitation of the study. In order to accommodate some variation in the direct and indirect impacts of mining operations the authors have included new analysis using a 10 km radius distance around the mapped mining sites. When the 10 km results are presented alongside the original 50 km analysis, some interesting new findings have emerged, and the original observations have held true. This has, therefore, been a worthwhile exercise that strengthened the paper.

I made another comment about coal being categorized as a non-renewable mining target in the original manuscript even though a portion of the world's mined coal is targeted for metallurgy, which then supports renewable energy infrastructure. The authors have clarified this in the revised manuscript and emphasize that coal mines belong to "other areas", with about 12% of the world's coal production being for metallurgical purposes, some of which may also be influenced by increasing demand for renewable energy infrastructure.

Reviewer #1

Thank you for an updated manuscript which has addressed the key points needing attention.

RESPONSE: We thank Reviewer 1 for their time and helpful suggestions, which improved the quality of our manuscript.

Reviewer #2

The authors have responded thoroughly to the comments provided in the first review and this has, in my opinion, improved the article. I agree with the central message that we need to plan future mining developments carefully during the transition to a low carbon economy and mitigate impacts on biodiversity as far as possible. However, I still have a couple of comments that I would like to see addressed that I hope increase the robustness of the findings. Most of these relate to the message that critical mines might have a disproportionately greater impact than other mining.

RESPONSE: We thank Reviewer 2 for their valuable comments and suggestions. Below we outline how our revisions address these additional comments.

1. L132 – 134: “but increasing the proportion of mines targeting materials critical for renewable energy production may also disproportionality increase the threats to biodiversity within currently protected areas.” I think this message probably needs to be expanded reflecting on the types of mines and how their impacts are managed given that the disproportionate impact is only based on the density of critical mining operations, a finding that is not robust to the impact radius of each mine. And a result that hasn’t been tested against some of the methodological assumptions – see below.

RESPONSE: We have added this important clarification to our manuscript (P3 L139–141). *“However, the ultimate impacts to biodiversity will depend on the mix of technologies used, their mineral needs and methods used to mine them, and the effectiveness of managing environmental impacts.”*

I am still concerned about two aspects of the analysis particularly given the slightly nuanced messages when comparing critical and other mining sites.

2. The first regards the sampling approach. The authors give no justification for the 100km cell interval, which is entirely arbitrary. Given that most of the findings comparing “critical” and “other mining” areas depend on this sampling, to what extent are the results dependent on this interval? To ensure that the results are robust, I would recommend that the authors do two things. First, justify why the sampling approach is needed, is this primarily to reduce data volume to make the analysis tractable? If it is removing autocorrelation because that breaks the assumption of data independence in the statistical tests, then the authors should show that the autocorrelation is removed at this scale or use a test that can account for spatial autocorrelation.

RESPONSE: We sampled our dataset to remove the autocorrelation created by our method, as clearly justified on P5P6 L271–273. *“We sampled mining and non-mining areas at 100-cell (~100 km) intervals (n=13,483), to remove potential issues with large sample sizes and the spatial autocorrelation created in our dataset by mapping mining areas using a 50-cell radius around each mining property.”*

Sampling at this distance also removed most other sources of spatial autocorrelation (now stated on P6 L273–279). *“Sampling at 100-cell intervals also removed most other sources of autocorrelation. We used GeoDa to produce correlograms⁴² quantifying correlation in mining density within 50-km*

distance bands (with a random sample of 50,000 cells and 1 million cell pairs). For mining areas created using a 50-cell radius, correlation by 100 km was 0.002 for areas targeting other minerals, 0.07 for areas targeting critical minerals, and 0.1 for both. For mining areas created using 10-cell radius, correlation at 100 km was 0.003 for others, 0.009 for critical, and 0 for both.” We have added these autocorrelation results from the spatial correlogram to the new Supplementary Table 7.

3. Second, would a different analysis distance or does the exact 100km grid impacts on the findings? The authors could test this by sampling at 100km (and another valid distance) using a set of different origins to generate a set of different samples and test that their findings are robust across these.

RESPONSE: To further investigate how sampling at 100 km intervals affected our results, we have now also sampled at 300 km intervals and repeated our analyses (P6 L287–291). *“Finally, to test the robustness of our results to the 100-cell sampling interval, we also sampled our datasets at 300-cell intervals (starting at a different point so that this sample was not simply a subset of the original sample) and repeated all analyses. We found relative differences between mining and non-mining areas and between critical and other mining areas do not change, although the significance of these differences sometimes do”.*

We have added these results to Supplementary Tables 3, 4 and 5 and updated our results accordingly. Specifically, we found: (1) the proportion overlap with KBAs for mining vs. non-mining, and with LoW for critical vs. other, became insignificant ($p=0.519$ and $p=3.20$, respectively) when sampling at 300 km intervals (P3 L112–115). However, the proportional overlap was still higher in non-mining regions than mining areas for KBAs and for other mining areas than critical areas for LoW. (2) The differences in density of critical vs. other mining areas overlapping with PAs and LoW became insignificant ($p=0.672$ and $p=0.175$, respectively) when sampling at 300 km intervals, although the mean density of overlapping areas were still higher for critical mining areas in both cases (P3 L131–133).

4. The second regards the buffering. Given the findings do show some sensitivity to the impact radius, does that not raise the question of how severe impacts are spread across the distance from the mine, in other words an impact decay kernel around each mine. I am not asking the authors to repeat their analysis with this approach as that would be a significant amount of work and the exact nature of such a kernel is uncertain, however, I think the manuscript would be improved if the discussion acknowledged some of the possible methodological issues associated with their analysis and possible avenues to improve this.

RESPONSE: Very few results reported in our manuscript differed when defining mining areas using a more the conservative 10-cell radius. In all cases, it was the significance level of the statistical test that changed, rather than the overall differences in proportional overlap or mean mining density (P3 L112–115, L132–133; Supplementary Tables 3–5). However, we do agree that a critical next step is to understand how indirect threats of mining to biodiversity differ spatially with increasing distance from mining properties. As suggested, we have added this opportunity for future research to our concluding paragraph (P4 L192–196): *“A systematic understanding of the consequences (rather than potential threats, as investigated here) of various mining activities on specific biodiversity features, including those that occur in marine systems and at varying distances from mine sites (rather than within a predefined distance of 50 km, as done here), is urgently required.”*

5. L52: “We distinguish areas targeting the materials critical for renewable energy production” could possibly more reasonably be reworded as “We distinguish areas targeting the materials

that amongst many other uses in the present day are also critical for renewable energy production” or words to that effect.

RESPONSE: We thank Reviewer 2 for this suggestion and have edited our manuscript accordingly (P2 L54–56): *“We distinguish areas targeting the materials that, among many other uses in society today, will also be critical for renewable energy production”.*

6. L211: “Counts represent mining density, and we converted density maps to binary values to indicate mining influenced areas.” For clarity it would be important to state what is left to be assumed here that this was equal to 1 where the density was greater than 0, and 0 when the density was equal to 0.

RESPONSE: We have clarified this point (P4 L219–220): *“Counts represent mining density, and we converted density maps to binary values to indicate mining influenced areas when mining density was greater than 0”.*

Reviewer #3 (Remarks to the Author):

1. I had only a few noteworthy concerns about the original manuscript, which have now been addressed to my satisfaction in the revised manuscript.

I suggested that the mapping approach would be more nuanced if different types of mines (in terms of targeted materials and associated mining methods) could be differentiated by their respective areas of impact. The authors have argued that this is beyond the scope of their work, and I do accept this limitation of the study. In order to accommodate some variation in the direct and indirect impacts of mining operations the authors have included new analysis using a 10 km radius distance around the mapped mining sites. When the 10 km results are presented alongside the original 50 km analysis, some interesting new findings have emerged, and the original observations have held true. This has, therefore, been a worthwhile exercise that strengthened the paper.

I made another comment about coal being categorized as a non-renewable mining target in the original manuscript even though a portion of the world’s mined coal is targeted for metallurgy, which then supports renewable energy infrastructure. The authors have clarified this in the revised manuscript and emphasize that coal mines belong to “other areas”, with about 12% of the world’s coal production being for metallurgical purposes, some of which may also be influenced by increasing demand for renewable energy infrastructure.

RESPONSE: We thank Reviewer 3 for their time and helpful suggestions, which improved the quality of our manuscript.

REVIEWERS' COMMENTS:

Reviewer #2 (Remarks to the Author):

Thank you for the revised manuscript, I think these amendments help to demonstrate some of the uncertainties around the central findings, which remain important.